# High-performance silk-based hybrid membranes employed for osmotic energy conversion

Weiwen Xin[1,2,5], Zhen Zhang[1,3,5], Xiaodong Huang[1], Yuhao Hu[1,2], Teng Zhou[4], Congcong Zhu[1,2], Xiang-Yu Kong [1], Lei Jiang[1,2] & Liping Wen[1,2]

The salinity gradient between seawater and river water is a clean energy source and an alternative solution for the increasing energy demands. A membrane-based reverse electrodialysis technique is a promising strategy to convert osmotic energy to electricity. To overcome the limits of traditional membranes with low efficiency and high resistance, nanofluidic is an emerging technique to promote osmotic energy harvesting. Here, we engineer a high-performance nanofluidic device with a hybrid membrane composed of a silk nanofibril membrane and an anodic aluminum oxide membrane. The silk nanofibril membrane, as a screening layer with condensed negative surface and nanochannels, dominates the ion transport; the anodic aluminum oxide membrane, as a supporting substrate, offers tunable channels and amphoteric groups. Thus, a nanofluidic membrane with asymmetric geometry and charge polarity is established, showing low resistance, high-performance energy conversion, and long-term stability. The system paves avenues for sustainable power generation, water purification, and desalination.

[1] CAS Key Laboratory of Bio-inspired Materials and Interfacial Science, Technical Institute of Physics and Chemistry, Chinese Academy of Sciences, Beijing 100190, P. R. China. [2] School of Future Technology, University of Chinese Academy of Sciences, Beijing 100049, P. R. China. [3] Beijing National Laboratory for Molecular Sciences (BNLMS), Key Laboratory of Green Printing, Institute of Chemistry, Chinese Academy of Sciences, Beijing 100190, P. R. China. [4] College of Mechanical and Electrical Engineering Hainan University Haikou, Hainan 570228, P. R. China. [5] These authors contributed equally: Weiwen Xin, Zhen Zhang. Correspondence and requests for materials should be addressed to X.-Y.K. (email: kongxiangyu@mail.ipc.ac.cn) or to L.W. (email: wen@mail.ipc.ac.cn)

Due to the ever-growing energy demands, osmotic energy as a clean and sustainable energy source has attracted significant attention from scientists[1,2]. With the development of nanofluidic systems and nanofluidic technology, the well-controlled ion transport in nanoconfined microenvironment brings novel insight for harvesting the osmotic energy, which is also called "blue energy"[3]. A promising and remarkable strategy, namely, reverse electrodialysis (RED), is used to harness the Gibbs free energy from natural waters[4–6]. Enormous scientific research toward membrane-based RED to obtain this worldwide energy has been extensively investigated[4–8]. A miniaturized device that directly generates electrical power from fresh and saline water with a power density of approximately 72 mW m$^{-2}$ was realized by utilizing ion exchange membranes[9]. In addition, by stacking multiple membranes and optimizing gap parameters, the power density of 2.2 W m$^{-2}$ can be obtained[10,11]. However, most ion exchange membranes used in traditional RED constrain the power density due to some limits, including inadequate mass transport and high membrane resistance[12–14]. Nanofluidic channel systems with controllable ion transport have been regarded as novel candidates to exploit the osmotic energy[15]. The enhanced ion selectivity and high mass flux endowed by the nanofluidics may greatly promote the osmotic power harvesting[16,17]. Siria et al. and Feng et al. successfully developed unique boron nitride nanotube and single-layer MoS$_2$ nanopores for osmotic energy harvesting and demonstrated the output power density of up to several 10$^3$ W m$^{-2}$ and 10$^6$ W m$^{-2}$, respectively[18,19]. These fundamental studies greatly stimulate the development of intelligent nanoporous membrane systems for various practical applications[20]. For example, a nanopower generator was designed by using a nanoporous polycarbonate track-etched membrane and achieved an energy density of ~0.058 W m$^{-2}$ [21]. To improve the energy density, our group has reported a series of composite systems based on track-etched membranes and self-assembly block copolymer (BCP) membranes, which achieved substantial increases in the output power density, up to 0.35 W m$^{-2}$ and 2.04 W m$^{-2}$, respectively[14,22]. However, these materials still suffer from high-cost, complex preparation, low output power density, and poor long-term stability, which constrain their practical applications[14]. Therefore, efforts devoted to pursue membrane materials with stability, high efficiency, and economically viable power density are urgently needed.

*Bombyx mori* (silkworm) silk, a most common natural biomaterial, owns outstanding mechanical properties, excellent biocompatibility, abundant surface groups, easy for chemical modifying, and large-scale cultivation[23,24]. Silk nanofibril (SNF) based materials have exhibited extensive applications in various fields, such as particle separation, flexible electronics, thermally insulating textiles, and wearable sensors[25–28]. Especially, because of the special molecular configuration of the silk fibroin, the abundant strong hydrogen bonds along with the van der Waals forces can form a thermodynamically stable structure[29], induced the excellent performance of SNF membrane in water purification[30]. On the other hand, anodic aluminum oxide (AAO) membrane is one of the most used substrate materials due to its controllable channel structures, accessible surface polarity tuning, and stability, which has been widely applied as templates and substrates in various fields, especially, in nanofluidic systems[31]. Thus, it would be interesting to couple the SNF and AAO materials for the ion transport investigation.

Here, we design a nanofluidic device prepared with a nanoporous SNF membrane and a variable-channel AAO membrane for osmotic energy harvesting. The SNF membrane with abundant negative surface and nanochannels dominates the ion transport; while the employed AAO substrate offers tunable channels and amphoteric groups, which established an asymmetric nanofluidic junction by combining with the SNF membrane in geometry and charge polarity. In view of the large number of β-sheets in SNF, numerous hydrogen bonds are formed on the interface between SNF and AAO, which contribute to the membrane's enhanced mechanical robustness and prominent aqueous stability[24]. It is worth noting that the proposed hybrid membrane can be easily fabricated without complex synthesis and thermal treatment, which are commonly needed for BCP-based and mesoporous carbon-based hybrid membranes[14,32]. Also, our fabricated membrane shows great long-term stability, and its power density exhibits no obvious decrement after three months (see Supplementary Fig. 1). The hybrid membrane shows typical surface-charge-governed ion transport behavior when the solution concentration is less than 1 M (KCl). The energy conversion performance of the hybrid membrane increases along with the pH changing, and power density get a maximum value of 2.86 W m$^{-2}$ by mixing artificial seawater and river water at basic condition. Furthermore, the performance of the energy harvesting under different concentration gradient is systematically investigated, including the AAO channel size screening and the SNF thickness changing. The established theoretical models and continuum simulation by employing PNP equations help us to understand how the AAO channel size, the SNF thickness, and the pH changing affect the ion transport and energy conversion, which could guide the future materials designing.

## Results

**Fabrication and characterization of hybrid membranes.** The native *B. mori* silk fibers are composed of silk fibroin coated with sericin proteins. To obtain the silk nanofibrils, the sericin proteins must be removed to deconstruct the hierarchical structure of silk fibers by using various solvent treatment[27]. Herein, the silk-based hybrid nanochannel membranes, SNF/AAO membranes, are fabricated with the assistant of vacuum filtration (Fig. 1a). The detailed three-step route for the membrane preparation is schematically shown in Supplementary Fig. 2 (texts in Methods). Scanning electron microscopy (SEM) and transmission electron microscopy (TEM) were employed to investigate the hierarchical structure of the silk fibers. Several silk fibers were linked together with the merit of sericin protein in natural silk (Fig. 1b). After the degumming treatment with the boiled NaHCO$_3$ solution annealing treatment, the cross-linked silk fibers were separated from each other, and the ambient sericin proteins were removed. Degummed silk fibers (Fig. 1c) were stable at room temperature for a long time[23]. After the incubation with 1,1,1,3,3,3-Hexafluoro-2-propanol (HFIP) and being sonicated (details in Methods), the silk nanofibrils were obtained and imaged with a TEM (Fig. 1d). The diameters of the SNF were ~20 nm, and the SNF showed a contour length within the range of 200 to 500 nm[33]. For the fabricated hybrid membrane, the cross-section SEM images revealed a 5-μm-thick SNF layer attaching onto a 60-μm-thick AAO substrate (Fig. 1e) whose structure was further examined (Supplementary Fig. 3a and b). The schematic illustration for the asymmetric nanofluidic structure was listed in Fig. 1f with a uniform SNF membrane surface (Fig. 1g). The inset shows the Tyndall light scattering of SNF solution, indicating the well dispersion of the silk fibrils.

The fabricated hybrid membranes demonstrated excellent water stability with being continuously soaked in deionized water for almost three months without decomposition and separation (Supplementary Fig. 1, Supplementary Fig. 3c and d). This preeminent stability is due to the strong interaction between the SNF and AAO membrane surfaces. The surface of SNF membrane existed abundant carboxyl, hydroxyl, and amino

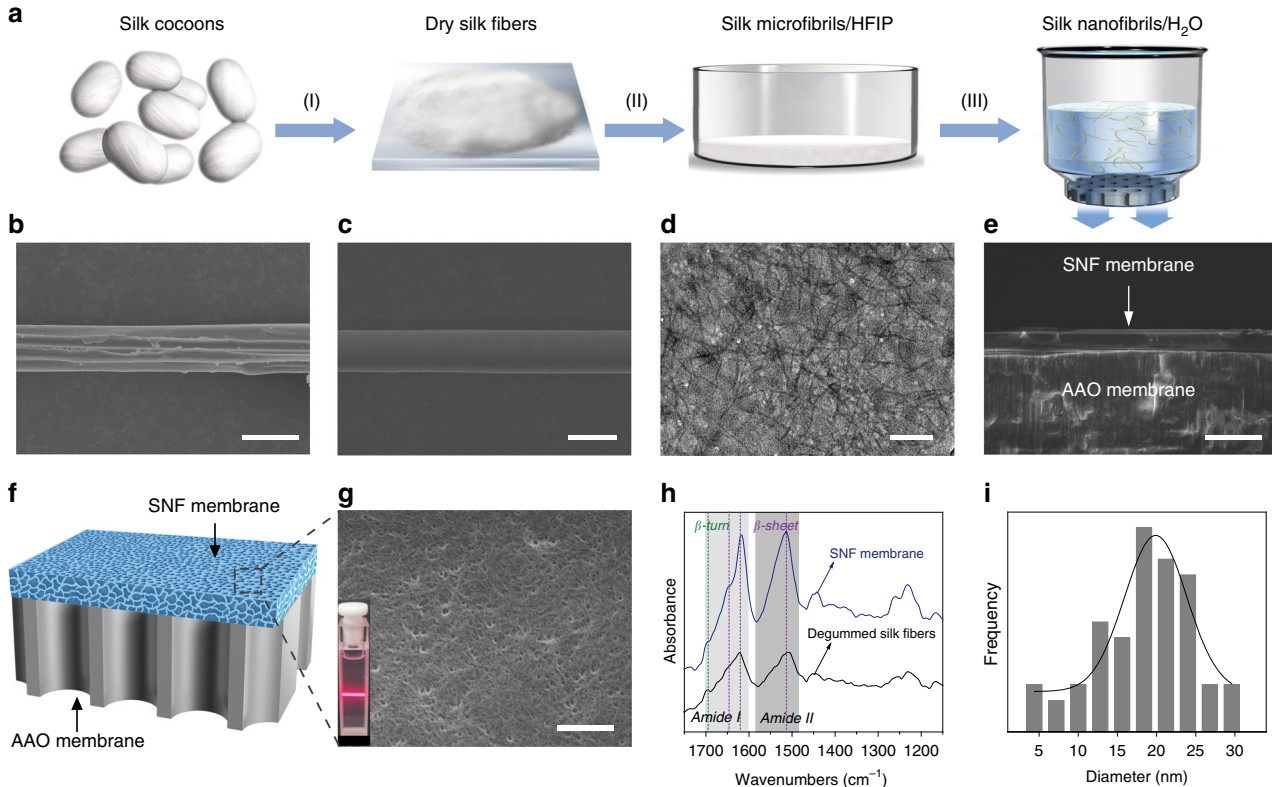

**Fig. 1** Fabrication and characterization of the nanochannel membranes. **a** Schematic fabricating process of silk nanofibril (SNF)/anodic aluminum oxide (AAO) membranes. (I) The silk cocoons are degummed to obtain the dry silk fibers. (II) Degummed silk fibers are mixed with 1,1,1,3,3,3-Hexafluoro-2-propanol (HFIP) to form slurries. (III) The SNF dispersion is assembled with AAO via vacuum filtration. **b** SEM image of a fiber composed of numerous concatenate silk fibers wrapped by vast sericin in natural silk cocoons. **c** SEM image of a degummed silk fiber. **d** TEM image of SNF, indicating a diameter of ~20 nm and a contour length within the range of 200–500 nm. **e** Cross-sectional SEM image of the hybrid membrane. The top layer is the SNF membrane, and the AAO membrane lies below. **f** A schematic illustration of the hybrid membrane. **g** SEM image of the SNF membrane surface. The inset shows the Tyndall light scattering of SNF solution. **h** FTIR spectra of the SNF membrane and degummed silk fibers. **i** The pore diameter distribution of the SNF membrane, and the average pore size is centered at ~20 nm. Scale bars: 20 μm (**b**), 2.5 μm (**c**), 0.2 μm (**d**), 10 μm (**e**), and 0.2 μm (**g**).

groups, and rich hydroxyl groups stood on the surface of AAO membrane (Supplementary Fig. 1b). Thus, a large number of hydrogen bonds could be formed between the SNF protein chains and –OH groups on AAO, including both the SNF|–O···H–O–| AAO hydrogen bond (21 kJ mol$^{-1}$) and the SNF|–N···H–O–|AAO (29 kJ mol$^{-1}$) hydrogen bond, respectively, which further enhanced the interface binding[34]. The molecular vibrational information of the SNF membrane was analyzed by employing the Fourier transform infrared spectroscopy (FTIR) in Fig. 1h. The strong peaks (1514, 1618, and 1647 cm$^{-1}$) and relative weak peak (1695 cm$^{-1}$) located in amide I and II band regions were assigned to the β-sheets and β-turns of the hairpin-folded antiparallel β-sheets structure[35,36], respectively, indicating that the SNF were mainly composed of β-sheets[26,30,31]. The schematic illustration of the hierarchical structure of the silk fiber showed the nanofibrils containing abundant β-sheets structure (Supplementary Fig. 4). The strong hydrogen bonds and van der Waals forces reinforced a thermodynamically stable structure[37]. Such structure endowed the excellent mechanical intensity of the membrane, and also accounted for the remarkable water stability which is similar to the natural silk fibers[29,38]. The pore diameter distribution of the SNF membrane was shown in Fig. 1i, and the average pore size centered at ~20 nm.

**Evaluation of electrochemical properties.** The ionic transport properties of the hybrid membrane were investigated with a homemade electrochemical device by monitoring the transmembrane ionic current (see Methods and Supplementary Fig. 5). To examine the stability of ion transport through hybrid membrane, the current-time (I–T) test of the hybrid system was conducted by alternatively applied an external bias of +1 V/−1 V. Each cycle lasted for 5 min without break and was repeated for a total period of 65 min. Figure 2a showed that both the negative and positive currents after each voltage polarity switch stayed at the same level over a wide pH range, indicating the excellent stability of the hybrid membrane. As shown in Fig. 2b, the transmembrane conductance remarkably deviated from the bulk value (black dash line) when the electrolyte concentration was <1 M, suggesting that the ion transport through the hybrid membranes was mainly governed by the surface charge[39]. The inset in Fig. 2b schematically showed that the channel surface of the SNF membrane held abundant carboxylic, amino and hydroxy functional groups[25]. We subsequently employed a separate AAO membrane and a separate SNF membrane (see Supplementary Fig. 6a–c for the assembling processes) for the energy conversion comparison with the hybrid membrane. The currents of the three membranes in the voltage ranged from 0 V to 1 V were recorded in 0.1 M KCl (pH ~6.80) (Fig. 2c). Benefiting from the macroporous structure that permitted ions to freely transport, the AAO membrane provided the maximum current values. The SNF membrane exhibited the minimum current values due to the nanoconfinement. In the meantime, the intermediate current of the hybrid membrane was benefiting from the introduction of the AAO channel with the ion-storage

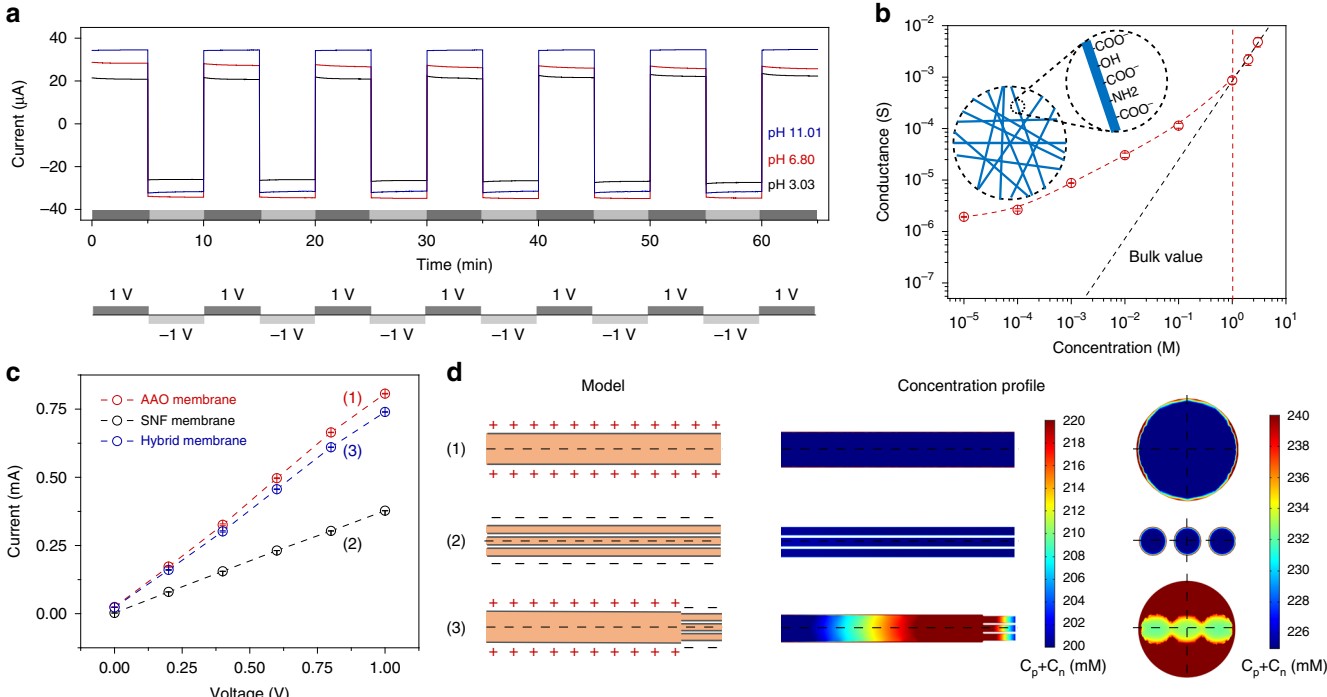

**Fig. 2** The stability and ion transport of the hybrid membranes. **a** *I–T* curves of the hybrid membrane recorded in 0.1 M KCl (pH 3.03, 6.80, and 11.01) with an external bias alternating between +1 V and −1 V to investigate the stability of ion transport through the hybrid membrane. Each cycle (+1 to −1 V) is sustained for 5 min. **b** Ionic conductance of the hybrid membrane as a function of the electrolyte concentration. The transmembrane ionic conductance (red circle) apparently deviates from the bulk value (black dash line) when the electrolyte concentration is less than 1 M, indicating the surface-charge-governed ion transport behavior. The inset shows the functional groups existing on the SNF membrane surface. Error bars represent s.d. **c** *I–V* curves of the AAO membrane, the SNF membrane, and the hybrid membrane measured with the voltage range (0 V, +1 V) in 0.1 M KCl. Error bars represent s.d. **d** The calculated ionic concentration profiles based on 3D models of the three membranes. The applied bias is set to +1 V

structure, which could largely induce the ion enrichment in the channels and showed the resultant bigger current than that of the SNF membrane. The ion distribution of the hybrid membrane was evaluated by employing the numerical simulation based on the Poisson and Nernst-Planck (PNP) equations[40] (see Methods and Supplementary Fig. 7). The ion concentration profiles based on 3D models calculated at +1 V bias of the three membranes were shown in Fig. 2d. The calculated ionic concentration profiles of the positively charged AAO (Fig. 2d, 1) and negatively charged SNF channel (Fig. 2d, 2) were both homogeneous. Also, the surface integral was employed for the comparison between the concentrations of AAO channels and SNF channels (see Numerical Simulation), and the calculated results agreed well with the experimental results (Fig. 2c). For the hybrid membrane, the ion concentration profile clearly showed a concentration gradient distribution (Fig. 2d, 3). The rectification of the hybrid membrane was due to the bipolar structure with the ion depletion and accumulation effects[12] at different voltage bias (Supplementary Fig. 8). Here, the bipolar structure could largely eliminate the concentration polarization which existed in the unipolar membrane[14,32]. Thus, the combined hybrid membranes gained the synergistic effect to simultaneously improve the ion transport and suppress the ion concentration polarization on the low-concentration side.

The high current and excellent ion-transmembrane-transport ability of the hybrid membrane paved the way for salinity gradient power harvesting. The hybrid membrane was placed between two-half electrochemical cells with different concentrations of NaCl solutions to extract the osmotic energy (Fig. 3a). Notably, ions in solution would diffuse from high-concentration side to low-concentration side through the ion-selective membrane, causing the directional flow of the cations or anions[41]. To

sustain the electroneutrality of the solutions, the electrode surfaces would undergo electrochemical redox reactions, and the electrons could be transferred to an external circuit for supplying an electrical device[1,16]. Thus, the hybrid membrane system could harvest part of the Gibbs free energy existing in the salinity gradient[42].

The energy conversion performance was investigated by recording *I–V* scans under series of salinity gradient. The open-circuit voltages ($V_{OC}$) and short-circuit currents ($I_{SC}$) could be obtained through reading the intercepts on the current and voltage axes by applying a sweeping voltage (−0.2 V, +0.2 V) at 20 mV steps. It was worth noting that the $V_{OC}$ was composed of two parts:[17,19] the diffusion potential ($E_{diff}$) generated by the power source and the redox potential ($E_{redox}$) generated by the unequal potential drop at the electrode-solution interface in different electrolyte concentrations (Supplementary Fig. 9). By applying the potential calibration (see Methods), the contribution of $E_{redox}$ was already deducted (Supplementary Table 1). Two configurations were arranged for salinity gradient harvesting: the concentrated solution on AAO side with $c_{AAO}/c_{SNF} = 0.5$ M/0.01 M and the concentrated solution on SNF side with $c_{AAO}/c_{SNF} = 0.01$ M/0.5 M (Fig. 3b). In the case of $c_{AAO}/c_{SNF} = 0.5$ M/0.01 M, the absolute values of $V_{OC}$ and $I_{SC}$ were 44 mV and 0.83 μA, respectively; while in the case of $c_{AAO}/c_{SNF} = 0.01$ M/0.5 M, both the $V_{OC}$ and $I_{SC}$ increased to 58 mV and 3.12 μA, respectively. The corresponding inner resistance ($r_{channel}$), which was calculated by $r_{channel} = V_{OC}/I_{SC}$, decreased by ~63%, which could promote the energy conversion in the configuration of concentrated solution on AAO side. In this configuration, the used electrolyte solutions (KCl) at the SNF side were fixed at 1 μM and those on the AAO side gradually increased from 1 μM to 3 M. The measured open-circuit voltages ($V_{OC}$) and short-circuit

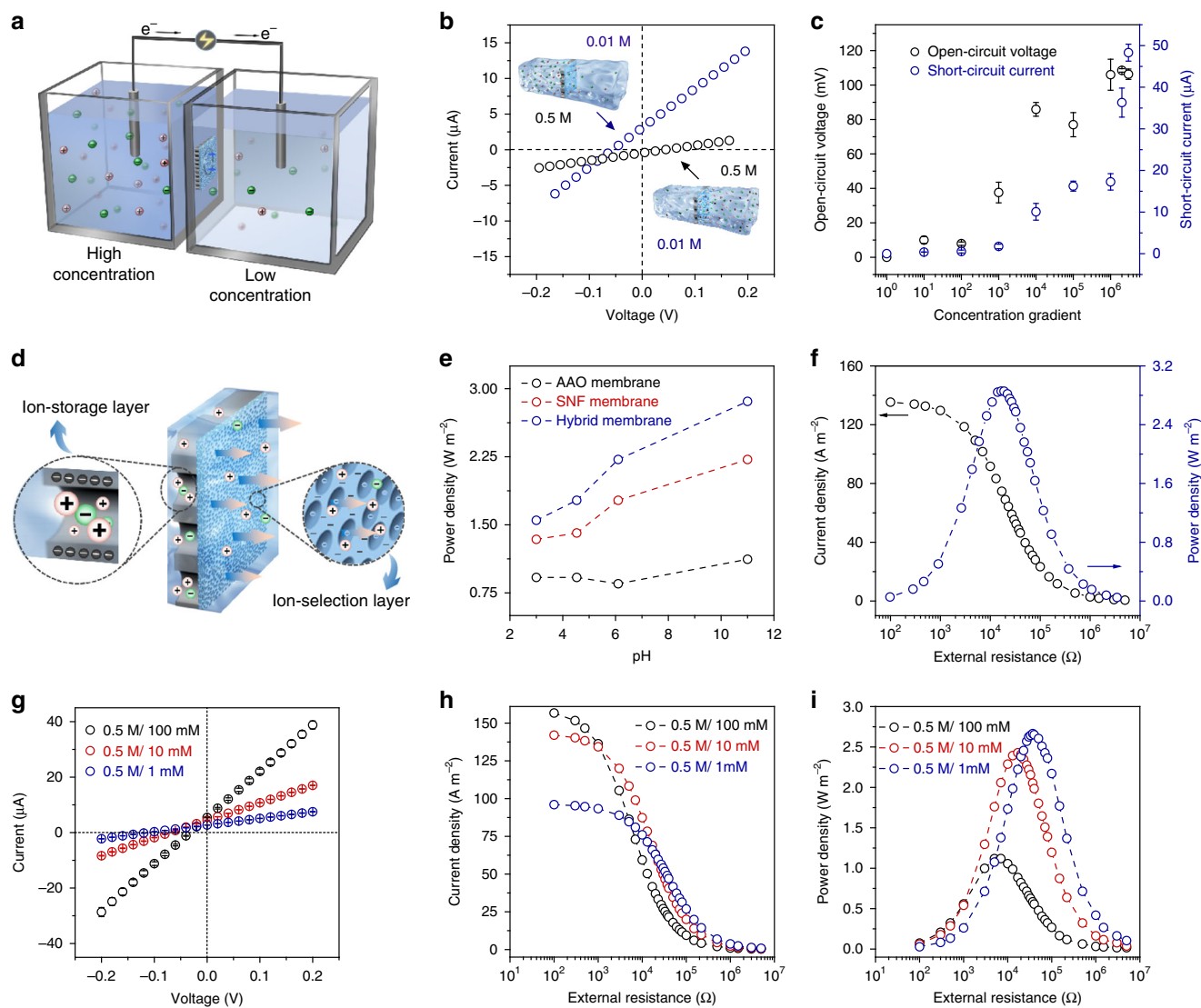

**Fig. 3** High-performance osmotic energy conversion. **a** Schematic of the energy harvesting device under a concentration gradient. **b** Two configurations for salinity gradient harvesting. **c** Measured $V_{OC}$ and $I_{SC}$ under different concentration gradient with the high concentration solution on AAO side. Error bars represent s.d. **d** Schematics of the AAO/SNF structure. The AAO and SNF membranes are considered to be an ion-storage layer and ion-selection layer, respectively. **e** The performance for energy conversion of the AAO, SNF, and hybrid membrane in different pH solutions. **f** The output power density and current density as the functions of load resistances. The output power density reaches a peak value of 2.86 W m$^{-2}$ at the load resistance of ~23 kΩ. **g** I–V curves under different salinity gradient. The high-salinity (NaCl) solution is placed on the AAO side and is fixed to 0.5 M. Error bars represent s.d. **h** Under three salinity gradients, the measured current densities all gradually decrease with the increasing external resistance. **i** The corresponding output power achieves the maximum values of 1.12, 2.43, and 2.67 W m$^{-2}$, respectively, for the 5-fold, 50-fold, and 500-fold salinity gradient

currents ($I_{SC}$) were shown in Fig. 3c with the peak values of approximately 125 mV and 48 μA, respectively. Moreover, the corresponding energy conversion efficiency sharply decreased from 30.1% to 5.5% as the concentration gradient increased from 10 to $3 \times 10^6$ (Supplementary Fig. 10). For illustrating the excellent ion transport promoted by the hybrid membrane, Fig. 3d showed the ion-storage and ion-selective functions offered by the AAO/SNF structure, in which the long AAO channels could store a great deal of ions and the negative narrow SNF channels functioned for the ions screening[32]. Therefore, the formation of the asymmetric AAO/SNF heterojunction resulted in the excellent performance for energy conversion, especially in the alkaline solutions (Fig. 3e).

The harvested power could be output to an external circuit to supply an electronic load resistor ($R_L$). In an artificial sea and river salinity gradient (50-fold concentration gradient with

0.5 M/0.01 M NaCl), the energy conversion performance of the hybrid membrane was investigated. The electric power density ($P_R$) was calculated according to the equation, $P_R = I^2 \times R_L$, where $I$ was the measured current under different concentration gradient. As shown in Fig. 3f, the diffusion current gradually decreased with the increase of load resistance, and the output power density reached its peak value when the load resistance was approximately 23 kΩ. The maximum power density was calculated to be 2.86 W m$^{-2}$. Also, the energy conversion performance under different concentration gradient was investigated by gradually increasing the concentration from 1 mM to 100 mM on the SNF side (Fig. 3g). Along with the concentration gradient increasing, the produced current density gradually increased (Fig. 3h). The corresponding energy conversion efficiencies for a 5-fold, 50-fold, and 500-fold concentration gradient were about 27.3%, 17.2%, and 12.9% with power density of 1.12, 2.43, and

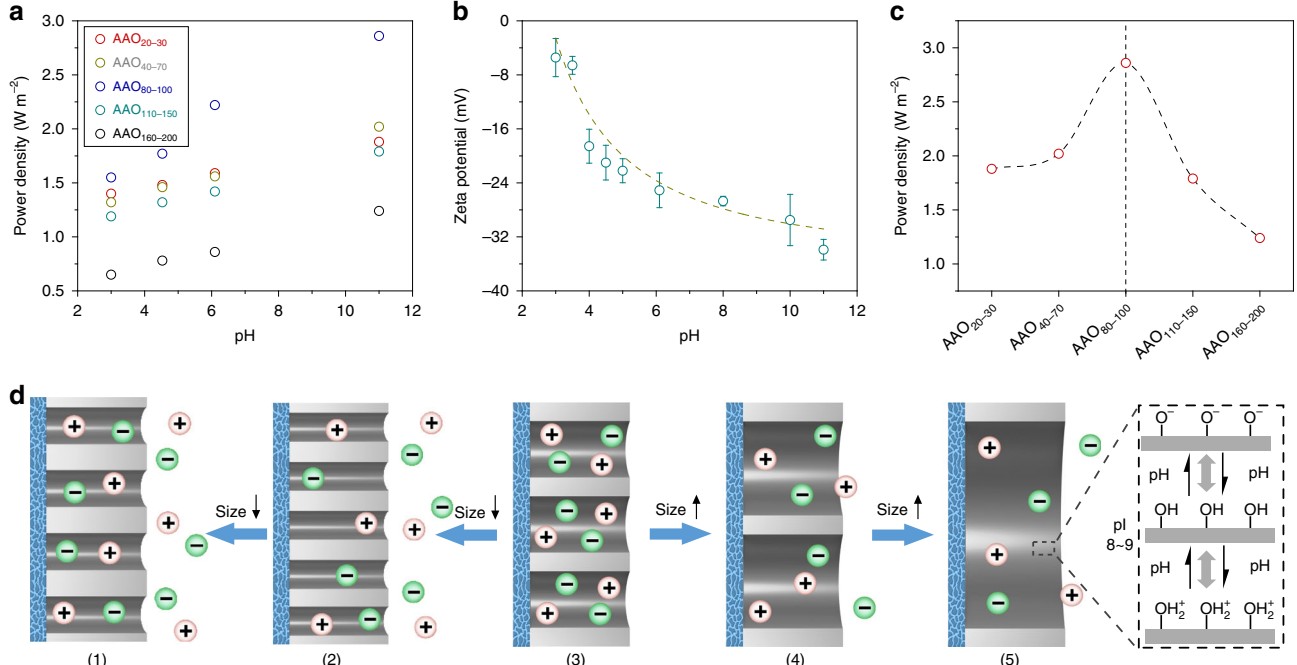

**Fig. 4** Effect of anodic aluminum oxide channel sizes. **a** The power density changes along with series of AAO channel sizes (from 20 to 200 nm) under different pH conditions. The thickness of the SNF membrane (10 μm) remains constant. **b** Zeta potential of SNF at pH from 3.00 to 11.00 in 0.01 M NaCl. Error bars represent s.d. **c** The power density of the hybrid membrane with the optimal channel size (80–100 nm) at pH 11.00 reaches 2.86 W m$^{-2}$. **d** Schematically illustrating the effect of AAO size on the energy conversion of hybrid membrane. AAO channel in Models (1): 20–30 nm, (2): 40–70 nm, (3): 80–100 nm, (4): 110–150 nm, (5): 160–200 nm, respectively. The right inset: the isoelectric point of alumina is ~8–9

2.67 W m$^{-2}$, respectively (Fig. 3i). Thus, as expected, the great energy conversion performance could be applied in high salinity gradient conditions.

**Effect of anodic aluminum oxide channel on energy conversion.** In order to investigate how the ion-storage channel affected on the energy conversion performance, the AAO channel size was screened at different pH values, and the optimum value was found to be in the range of 80 to 100 nm (Fig. 4a). As the pH increased from 3.00 to 11.00, the power density increased for all sizes, indicating the hybrid membrane would show better performance in alkaline environment. Also, the measured zeta potential of SNF gradually increased as the pH increase (Fig. 4b), indicating a cation-selective membrane[26]. On the other hand, the surface polarity of AAO channels was pH-dependent because of the amphoteric -OH groups. Thus, the hybrid channels were all negatively charged at pH 11.00, which facilitated the transport of cations and promoted the energy conversion with a peak value of 2.86 W m$^{-2}$ (Fig. 4c, Supplementary Fig. 11).

To screen the AAO channel size, we employed a series of AAO membranes with increasing channel sizes (from 20 to 200 nm) and proposed the corresponding models in Fig. 4d. For the AAO channel size less than 70 nm, Model (1) and (2), the ion-storage function was limited due to the effective contact of between SNF channels and AAO channels obviously decreased which resulted in the damage of ion-transmembrane transport (Supplementary Fig. 12). When the AAO channels were the range of 20–30 nm, the storage space was further compressed. More importantly, the size of SNF and AAO channels was similar, equivalent to generating much longer path resistance. When the channel size increased to the range of 80–100 nm, Model (3), sufficient ions entered into the channel and realized the ion-storage function. The continuously increasing channel size to exceeded 110 nm, Models (4) and (5), weakened the ion-storage function of AAO channel, and in a tense, eliminated the formed heterogeneous

junction with SNF membrane. The optimized AAO channel size would benefit the hybrid membrane in the osmotic energy conversion.

**The reversal of optimal silk nanofibril membrane thickness.** The thickness of the SNF membrane affected the energy conversion of the hybrid membrane largely. Hybrid membranes with the SNF membrane thicknesses from 5 μm to 80 μm were controlled by adjusting the volume of the exfoliated SNF dispersion for filtration (Supplementary Fig. 13). Regardless of the SNF membrane thickness changing, the energy conversion increased along with the pH increasing (Supplementary Fig. 14a). As shown in Fig. 5a, the optimal thicknesses of SNF membrane at pH 3.00 and 11.00 for energy conversion were in the range between 15 μm and 10 μm. Thus, the power density changes of these two membranes along with the pH increasing were investigated (Fig. 5b). In Part I with the pH < 4.53, the membrane with thicker SNF membrane showed higher power density. Interestingly, in Part II, the two membranes got closed power densities when the pH increased to ~4.53, which was called as "turning point" (Fig. 5b). In Part III with pH > 4.53, the power densities of the two membranes both increased and the membrane with thinner SNF membrane showed excellent energy conversion performance. These results indicated the trade-off relation between the SNF membrane thickness and pH values. Herein, we proposed the models in Fig. 5c for the illustration of this phenomenon. The three parts in Fig. 5c denoted as gray Part I, blue Part II, and light-green Part III for the $P_{15} > P_{10}$, $P_{15} \approx P_{10}$, and $P_{15} < P_{10}$ (P: power density; subscript "15" and "10": the thickness of the SNF membrane) parts in Fig. 5b, respectively. The Debye-Hückel equation was used to describe Debye length:[43]

$$\lambda_D = \sqrt{\frac{\varepsilon \varepsilon_0 RT}{2n_{bulk} z^2 F^2}} \quad (1)$$

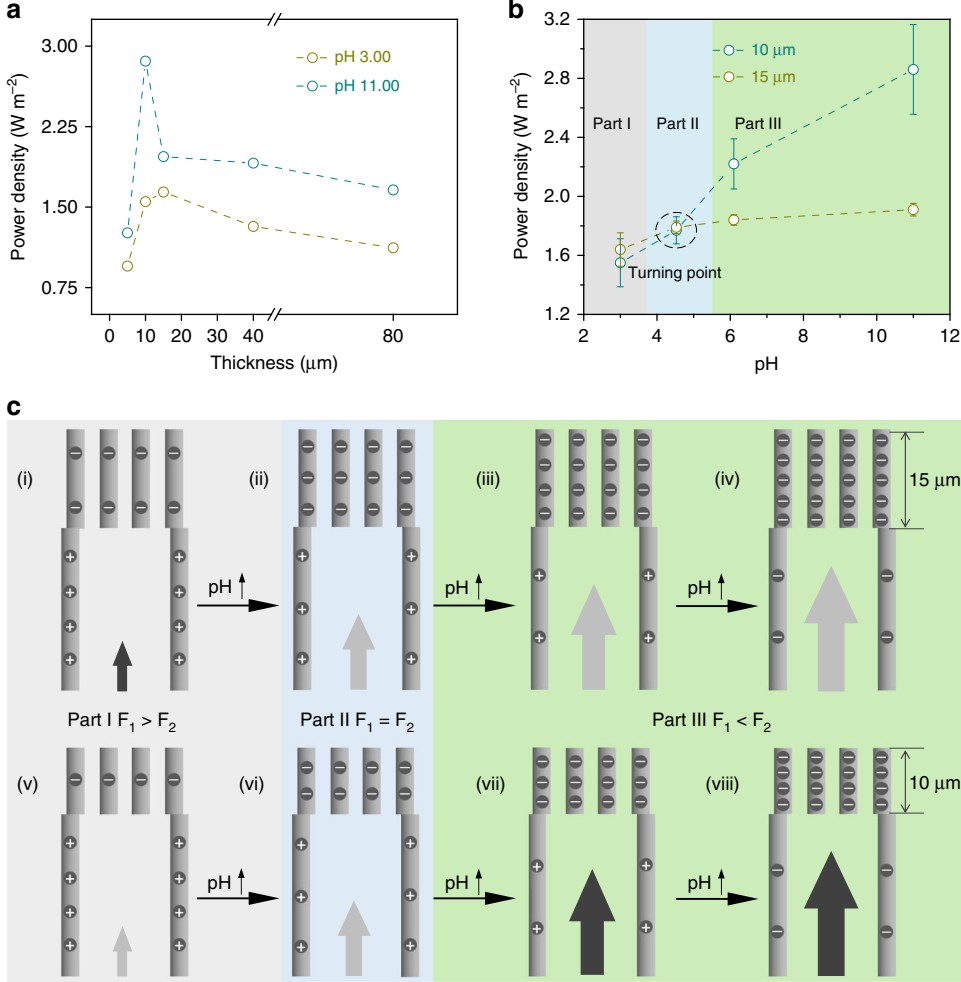

**Fig. 5** The effect of silk nanofibril membrane thickness **a** Altering the thickness of SNF membranes. The curves of the power densities of the hybrid membranes and the SNF membrane thickness reflect a trend that first increase and then decrease at pH 3.00 and 11.00. **b** The turning point of the optimum thickness occurs at pH ~4.53. Three regions are employed to distinguish the different situations, including $P_{15} > P_{10}$, defined as Part I; $P_{15} \approx P_{10}$, defined as Part II; and $P_{15} < P_{10}$, defined as Part III (P represents power density; corner mark (15 or 10) indicates the thickness of the SNF membrane). Error bars represent s.d. **c** Schematic of the ion-transport ability and the effects of different forces. Schematics (i)–(viii) of the charge distribution on the internal surface of AAO channels under different pH conditions. There are two kinds of forces acting on ions in hybrid membranes, including the path resistance force (defined as $F_1$) and the driving force (defined as $F_2$). Part I indicates that $F_1$ is more robust than $F_2$; oppositely, $F_1$ is weaker than $F_2$ in Part III. The light blue area called Part II expresses that the two forces are equivalent. The background color is the same as that in Fig. 5b and stands for the magnitude of the power density. The arrows are used as a proxy for the resultant force under the corresponding conditions. The sizes of the arrows signify the magnitude of the resultant force. For arrows with the same size, the darker the color is, the larger the resultant force is

where $\varepsilon$, $\varepsilon_0$, $n_{bulk}$, and $z$ were the permittivity of water, the permittivity of a vacuum, the concentration of solution, the valence number, respectively. $F$, $T$, and $R$ represented the Faraday's constant, the absolute temperature, and the universal gas constant, respectively. Hence, the calculated Debye length ($\lambda_D$) in 0.01 M NaCl was ~3.1 nm, which was far less than half of the AAO channel size (~40 nm), showing little electric double layer screening effect. In contrast, the $\lambda_D$ was close to half of the SNF channel size (~10 nm), and the ion transport could be affected.

There were two main forces applied on the ions in the hybrid membranes, the resistance force ($F_1$) and the driving force ($F_2$). The $F_1$, which restricted effective ion transport, could be illustrated by

$$F_1 = \int P_r dl + F_c(l) + F_f(l) \tag{2}$$

where $P_r$ and $l$ represented the vapor pressure and the thickness of the SNF membrane, respectively. Both $F_c(l)$ and $F_f(l)$ were functions of $l$ that expressed ion collision and friction,

respectively. The effect of $F_1$ was mainly sourced from the SNF channel. The $F_2$, that promoted the ion transport, could be expressed by

$$F_2 = \int P_O dl \tag{3}$$

where $P_O$ was composed of the osmotic pressure and Maxwell stress[44], which could be expressed by

$$P_O = n_{bulk}kT\left(\frac{zeV}{kT}\right)^2 \frac{2\cosh 2\kappa x}{\cosh 2\kappa h + 1} + 2n_{bulk}kT \tag{4}$$

where $n_{bulk}$, $k$, $T$, and $\kappa$ were the concentration of the electrolyte (NaCl), Boltzmann's constant, the absolute temperature (here, 298.15 K), the inverse of Debye length, respectively. $V$ was a function of the surface charge density $\sigma$ and could be calculated by

$$V = \frac{\sigma}{\kappa\varepsilon\varepsilon_0} \frac{\cosh \kappa h}{\sinh \kappa h} \tag{5}$$

thus,

$$F_2 \propto F(\sigma, l) \qquad (6)$$

According to equation (6), along with the pH increasing, the surface charge density ($\sigma$) and the thickness of SNF membrane ($l$) increased (Fig. 5c). Taking the situation at the "turning point", the $F_1$ and $F_2$ were set equal for assisting the analysis. The relative magnitude of the two forces was given in the middle of Fig. 5c. The arrows denoted the power density of two hybrid membranes. The sizes of the arrows were used for the comparison with the same membrane at different pH, and the colors of the arrows were used for the comparison with the two membranes at the same pH, respectively. For the top row in Fig. 5c, along with the pH increase, the $\sigma$ gradually increased from (i) to (iv), and the driving force, $F_2$, increased accordingly. Also, due to the low surface charge density in acidic environment, $F_2$ in (i) presented less than the resistance force $F_1$, showing relative low power density. From Part I to Part II and Part III, the $F_2$ increased and gradually surpassed $F_1$, promoting the ion transport and resulting in the high power density. It was the same situation for the hybrid membrane with 10 μm SNF membrane. For the (i) and (v) in Part I, due to the thicker SNF layer, the cation transport in the negatively charged channels (i) could be promoted and showed higher power density than that in (v). Yet, in Part III, the increased $\sigma$ coupling with the reduced $F_1$ induced the promotion of the ion transport in thinner SNF membrane, (vii) and (viii), and superior power density than those in (iii) and (iv), respectively. Also, the numerical simulations based on PNP theory agreed well with the experimental results (Supplementary Fig. 14b).

## Discussion

In summary, we have demonstrated that the design of applying a silk-based hybrid membrane for RED device is a promising avenue for osmotic energy harvesting. The hybrid membrane with heterogeneous geometry, chemistry, and electrostatic potential is proved able to promote the ion transport. The thickness of the selective-function SNF layer and the size of the ion-storage AAO channel are screened to maximize the energy conversion, and the power density of 2.86 W m$^{-2}$ can be achieved at 50-fold concentration gradient. Also, the proposed membrane shows the capability of broad working environment in wide pH range. Especially in alkaline solutions, the hybrid membrane demonstrates excellent energy conversion performance, giving the clue of its broad prospects in collecting energy from various waters (Supplementary Table 2). Significantly, benefiting from the β-sheet in silk fibroin and formed abundant hydrogen bonds in the SNF/AAO interface, the hybrid membrane exhibits outstanding long-term stability. Also, due to the large-scale cultivation of silkworms, the used natural silk materials hold abundant and low-cost sources for the practical application of the proposed system. Thus, the introducing of silk-based materials and controlled ion transport in nanoconfined environment uncovers the great potential for the sustainable osmotic energy harvesting.

## Methods

**Chemicals.** *Bombyx mori* cocoons were collected from Shanghai Buke Co., China; 1,1,1,3,3,3-Hexafluoro-2-propanol, HFIP (purity: ≥ 99%) was bought from Sigma-Aldrich, USA; The anodic aluminum oxide (AAO) membranes were purchased from Hefei Pu-Yuan Nano Technology, Ltd., China; Other chemicals were all analytical grade and obtained from Sinopharm Chemical Reagent Co. Ltd., China. All solutions were prepared using Milli-Q water (18.2 MΩ cm).

**Exfoliated SNF dispersion.** Silk fibroin aqueous solution was prepared from *Bombyx mori* silkworm cocoons by successive procedures (Supplementary Fig. 2) including cutting up, degumming, dissolving, dispersing, and filtrating as described in literatures[18,24]. In detail, 0.5 wt% sodium bicarbonate (NaHCO₃) solution was

prepared by dissolving 10.00 g of NaHCO₃ in 2-l of deionized water and heated until boiling. *Bombyx mori* cocoon was cut into finger-shaped pieces and degummed twice in the boiled NaHCO₃ solution for 30 min. Then the degummed silk fibers were thoroughly washed for three times with the cold deionized water. Then, the silks were spread on a new piece of aluminum foil and dried in air at room temperature overnight. The well mixed degummed silk fibers/HFIP mixture with a weight ratio of 1:30 was incubated at 60°C for 24 h with airtight container. Then, the silk microfibrils pulp was dried in a fuming cupboard to evaporate HFIP for approximately 5 h, followed by adding deionized water with a weight ratio of 1:400 under continuous stirring. The undissolved materials were removed manually. Finally, the microfibrils/water mixture was sonicated at 40 kHz frequency for 60 min, the exfoliated SNF dispersion (0.05 wt%) was collected by centrifugation at 10000 rpm for 30 min.

**Fabrication and characterization of hybrid membrane.** The SNF/AAO membrane was prepared via vacuum filtration. In brief, the SNF dispersions are vacuum-filtrated onto AAO membrane with different pore size through a vacuum filtration device. The thicknesses of the SNF membranes could be controlled by the volumes of the SNF dispersions with certain concentration. A series of characterizations were conducted for the hybrid membrane. Contact angle measurement was tested using an OCA25LHT (Datephysics, Germany) instrument. FTIR spectra was taken by using a Fourier transform infrared spectroscopy (Excalibur 3100 spectrometer, Varian, USA) to investigate the secondary structures of proteins by investigating the amide vibrations. The testing area of the SNF membrane is about 1 cm² and the degummed silk fiber sample was weighted to be 0.05 g. SEM images were recorded by using a scanning electron microscopy (Hitachi S-4800, Hitachi, Japan) instrument with the acceleration voltage of 10 kV. TEM images were taken using a transmission electron microscopy (JEOL, JEM-2100, Japan).

**Electrical measurements.** The ionic transport properties and the energy conversion tests of the hybrid membranes were measured by using a Keithley 6487 semiconductor picoammeter (Keithley Instruments, Cleveland, OH). In brief, the hybrid membranes were mounted between a two-chambers of the testing cell, and the unique design provides thick feed-water troughs (Supplementary Fig. 5) which are desired in electrodialysis to reduce salt depletion in the boundary layers adjacent to the membranes[3,11]. The redox potential is obtained by measuring the potential difference without the hybrid membrane. In all the measurements, the effective experimental area is about $3 \times 10^{-2}$ mm². Homemade Ag/AgCl electrodes are used to apply a transmembrane potential across the membrane. The anode was set on the SNF membrane side. For the I–V curves of the AAO membrane, the SNF membrane, and the hybrid membrane, were measured with the voltage range (0 V, + 1 V), and the bias of +1 V was set on AAO side. The pH values of the used electrolytes (KCl or NaCl) solutions were adjusted by using corresponding acidic or alkaline solution (HCl/KOH/NaOH). All the measurements in the current work except the current-time testing are accomplished within 3 min in order to minimize the influence of CO₂ absorption (Supplementary Fig. 15). The testing electrolyte solutions are always refreshed before each measurement. Additionally, the resistance analysis of the system was conducted to evaluate the system's performance[45]. The equivalent circuit was shown in Supplementary Fig. 16a; the resistance of the solution at different concentration was listed in Supplementary Fig. 16b; the comparison of the total resistance ($R_t$) and total resistance of multiple components in the circuit excluding the membrane resistance ($R_m$) at different concentration gradients was listed in Supplementary Fig. 16c, respectively. Along with the concentration gradient increasing, both the $R_t$ and the $R_m$ gradually decreased, and the $R_t$ was mainly contributed by $R_m$. Thus, by bringing down the $R_m$, the output power density could be further increased.

**Numerical simulation.** Numerical simulation was performed using a commercial finite-element software package COMSOL (version 5.4) Multiphysics. The Poisson and Nernst-Planck (PNP) equations are shown below:[40]

$$j_i = D(\nabla c_i + \frac{z_i F c_i}{RT} \nabla \varphi) \qquad (7)$$

$$\nabla^2 \varphi = -\frac{F}{\varepsilon} \sum z_i c_i \qquad (8)$$

$$\nabla \cdot j_i = 0 \qquad (9)$$

where, $j_i$, $D_i$, $c_i$, $z_i$, $\varphi$, and $\varepsilon$ are the ionic flux, diffusion coefficient, ion concentration, valence number for each species $i$, electrical potential, and dielectric constant of the electrolyte solution, respectively. $F$, $R$, and $T$ are the Faraday constant, universal gas constant, and absolute temperature, respectively. Equation (7) is the flux equation for each ionic species (Nernst-Planck equation) which physically describes the transport properties of a charged nanopore. Equation (8) is the Poisson equation which describes the relationship between the electrical potential and ion concentrations. The system is generally simplified by assuming steady-state conditions, and the flux should satisfy the time-independent continuity Eq. (9) when the system reaches a stationary regime. With the given geometry and appropriate boundary conditions, the coupled Eqs (7)–(9) could be solved with finite-element calculations for the ion concentration distribution. The numerical

simulated model based on 3D structure is shown in Supplementary Fig. 7. It contains a 1000 nm long AAO channel (pore size: 80 nm) and a 100 nm long SNF channel (pore size: 20 nm), which is consistent with the experimental geometry. Two electrolyte reservoirs are introduced to decrease the effect of entrance/exit mass transfer resistances on the overall ionic transport. The external voltage is applied on the boundary $W_1$ and the wall $W_2$ offers the reference potential. The ion flux has the zero normal components at boundaries:

$$n \cdot j_i = 0 \tag{10}$$

The boundary condition for the potential $\varphi$ on the channel walls is:

$$n \cdot \nabla \varphi = -\frac{\sigma}{\varepsilon} \tag{11}$$

where, $\sigma$ represents the surface charge density. Notably, the surface charge density of the AAO channel and the SNF channel are set to $+0.08$ C m$^{-2}$ [22,32] and $-0.006$ C m$^{-2}$, respectively. The charge density of SNF was calculated according to the equation:[46]

$$\sigma = \frac{\varepsilon \varepsilon_0 \xi}{\lambda_D} \tag{12}$$

where $\varepsilon$, $\varepsilon_0$, $n_{\text{bulk}}$, and $z$ were the permittivity of water, the permittivity of a vacuum, the concentration of solution, the valence number, respectively. $F$, $T$, $R$, and $\zeta$ represented the Faraday's constant, the absolute temperature, the universal gas constant, and zeta potential of SNF ($-8.59$ mV in 0.1 M KCl at room temperature), respectively. Debye length ($\lambda_D$) can be obtained according equation (1). Also, the length ratios of the AAO/SNF channels are gradually decreased from 12 to 0.75 to investigate the effects of SNF channel lengths on the whole concentration profile (Supplementary Fig. 14b). The surface integral was employed to compared the concentrations of AAO channels with SNF channels in the simulation models. The calculated values of $c_{AAO}$ and $c_{SNF}$ are $4.0028 \times 10^{-12}$ and $7.4601 \times 10^{-13}$ mol m$^{-1}$, respectively, which agreed well with the results in Fig. 2c.

**Potential calibration.** The energy conversion properties of the hybrid membrane are studied by applying sweeping voltages. As the power source, the Supplementary Fig. 9 indicates the equivalent circuit of the membrane under a concentration gradient. $r_{\text{channel}}$ represents the internal resistance of the hybrid membrane. $V_{OC}$, $E_{\text{diff}}$ and $E_{\text{redox}}$ represent the measured open-circuit voltage, the diffusion potential of the hybrid membrane, and the redox potential. The measured open-circuit voltage ($V_{OC}$) actually consists of two parts: the diffusion potential ($E_{\text{diff}}$) that is generated by the power source and the redox potential that is produced by the unequal potential drop at the electrode-solution interface in different electrolyte concentrations, which satisfy the following equation:

$$V_{OC} = E_{\text{diff}} + E_{\text{redox}} \tag{13}$$

thus,

$$E_{\text{diff}} = V_{OC} - E_{\text{redox}} \tag{14}$$

For the electrode calibration, the $I$–$V$ curves were recorded by applying a sweeping voltage ($-0.2$ V, $+0.2$ V) at 20 mV steps with and without the hybrid membrane, respectively. In the case of with no hybrid membrane, the measured voltage was determined individually by the $E_{\text{redox}}$, and the values of $E_{\text{diff}}$ could be obtained (Supplementary Table 1).

**Energy conversion efficiency.** Normally, the energy conversion efficiency corresponding to the maximum power generation ($\eta_{\text{max}}$) in the system can be calculated by the equation:[13,38]

$$\eta_{max} = \frac{1}{2}(2t_n - 1)^2 \tag{15}$$

where $t_n$ represent the anion transference number. Here, when the membrane material is perfectly cation selective, $t_n$ reaches the max of 1. The maximum efficiency evaluated by the equation is 50%. Hence, the value of tn can be obtain:

$$t_n = \frac{1}{2}\left(\frac{E_{\text{diff}}}{\frac{RT}{zF}\ln\left(\frac{\gamma_{c_H}c_H}{\gamma_{c_L}c_L}\right)} + 1\right) \tag{16}$$

where $E_{\text{diff}}$, $R$, $T$, $F$, $z$, $\gamma$, and $c$ refer to the diffusion potential, universal gas constant, the absolute temperature, Faraday constant, charge number, activity coefficient of ions, and ion concentration, respectively. From Eqs. (15) and (16), it is concluded that the $\eta_{max}$ is proportional to $t_n$. Here, the $t_n$ increases with a decrease of the concentration of either side in the nano-sized channel[17]. Therefore, along with the concentration increasing, the $t_n$ decreases, leading to the $\eta_{\text{max}}$ decreases. For example, the energy conversion efficiency for the energy harvesting system employed in five-fold, 50-fold, and 500-fold concentration gradients is about 27.3%, 17.2%, and 12.9%, respectively. Additional data under various concentration gradients can be calculated and listed in Supplementary Fig. 10.

## Data availability
The data that support the findings of this study are available from the corresponding author upon request.

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

## Acknowledgements
This work was supported by the National Key R&D Program of China (2017YFA0206904, 2017YFA0206900), the National Natural Science Foundation of China (21625303, 51673206, 21434003), the Strategic Priority Research Program of the Chinese Academy of Sciences (XDA2010213), Beijing Natural Science Foundation (2194088), Beijing Municipal Science & Technology Commision No. Z181100004418013, and the Key Research Program of the Chinese Academy of Sciences (QYZDY-SSW-SLH014).

## Author contributions
L.W. proposed the research direction and guided the project. W.X. and Z.Z. designed and performed the experiments. L.W., X-Y.K., W.X., Z.Z., and L.J. analyzed and discussed the experimental results and drafted the paper. W.X. and T.Z. performed the numerical simulations. X.H., Y.H., and C.Z. joined the discussion of data and gave useful suggestions. All authors contributed to the writing of the paper.

## Additional information

**Competing interests:** The authors declare no competing interests.

