## [Peer Review File · Nature Communications]

Reviewers' comments:

Reviewer #1 (Remarks to the Author):

This is a nice manuscript reporting the application of silk nanofiber-based membranes for osmotic energy conversion. The porous charged silk membrane standing on anodic alumina oxide functioned as the junction for selective ion transport between concentrated and less concentrated water, converting thermodynamic salinity gradient into energy. The topic of this manuscript is timely and interesting, shows the new way of harvesting energy with easily available ion-selective membrane. The membrane, in comparison with others studied previously by others or the same group, is ease of preparation and highly stable, with possibility of large-scale production. The performance of this osmotic energy conversion system bring a significant improvement with a maximum power density up to 2.86 W/m². The authors also carried out a complete study with both experimental and simulations depth. This work is publishable by the journal, but only after the authors fully address the following comments.

1. I do have a serious reservation on the model of numerical simulation the authors used, which I believe is completely incorrect. The silk fiber membrane consists of tortuous pores or channels but considered as straight, this is not scientifically sound or reliable. Even though this simplification is acceptable, the 2D-plane model should be definitely replaced with 2D-axisymmetric model or 3D model. In 2D-plane model, the model geometry is not a cylinder, but a cuboid with a length of 1 m (the default parameter of COMSOL software). The use of correct model is compulsory, and then compare the correct simulation with experiment.
2. One of key issues in this system is how to balance the ion selectivity and mass flux, in which surface charges play the decisive role. The authors assigned the charge density to be 0.024 C/m². However, I did not find the relevant characterization of surface charge state or how they got this value. The value is also very important for simulation.
3. The authors should describe much more details of experiments. For instances, how they measure Eredox, and specify clearly the experimental membrane area in this energy conversion system, as well as the area they used for estimating the conversion efficiency.
4. Line 224, "The corresponding inner resistance (rchannel), which was calculated by $r_{channel} = VOC/ISC \dots$ ". This resistance also includes the solution resistance.

Reviewer #2 (Remarks to the Author):

The manuscript from Wen et al. presented the fabrication and the performance of a silk-AAO hybrid membrane system for harvesting osmotic energy. The membrane was prepared through easy filtrating silk nanofibers with alumina membranes and presented excellent energy conversion performance. The energy conversion of the hybrid system was investigated in detail by considering a series of factors, such as ionic concentration gradients, the opening diameter of the alumina part, and thickness of the silk part of the system, which provided systematical guidance for membrane design and speeding up the development of the current filed. Also, the factor analysis was clearly conducted by employing the theoretical simulation. The presented system was scalable, easily accessible, and of potential interdisciplinary interests. The data is solid and well presented with high-quality pictures. In the meantime, this work will definitely arouse the interest of the membrane-based energy conversion with a series of low-cost materials and draw much attention from not only the academic but also the industrial community. Overall, I would like to recommend this work to be published in Nature Communications after the authors consider the following minor suggestions.

1. Normally, the resistance of the membrane for RED is a key factor for energy conversion performance. The current membrane showed low resistance which could benefit the energy conversion process. The fig. 3i shows the resistance increase as the concentration gradient increasing and supplementary fig. 10 shows the energy conversion efficiency decreased with the salinity gradient increasing. Thus, the authors are recommended to discuss how the concentration gradient affected the

output power performance in detail for readers' interest.

2. For the AAO channel size screening part, the authors used the effective pore overlapping for analyzing the output power generation. Here, the AAO layer and silk layer are simplified to round pores with the proper size from experimental results. It's an interesting method for the discussion and easy to understand. I have noticed that the randomly generated silk layer pores distributed differently in supplementary figure 12a and 12b. Thus, I wonder how the authors generate randomly pore distribution? Also, should the generated pore distribution different in the comparison or might the authors employ more generated random silk layer pore arrays for the analysis?

3. In the manuscript, the authors adjusted the pH of the solution for the RED use. As known that the energy conversion is benefited from SNF/AAO heterogeneous structure, especially the opposite charge polarity in the alkaline condition. Thus, could the authors comment on the potential methods to adjust the optimal membrane for certain applications or the direction of the heterogeneous membrane development.

4. For the practical applications, the provided data table in supplementary materials helps a lot for readers' information. Also, the analysis related to fig. 4 and fig. 5 may help promote the performance of the system in practical use. The authors are recommended to comment on the practical use of the current system and the concerns or factors for developing future systems.

Reviewer #3 (Remarks to the Author):

The authors investigated silk-based hybrid membranes for reverse electrodialysis. This paper showed interesting results regarding ion transport phenomena in hybrid membranes for reverse electrodialysis. However, I believe that the quality of the paper can be further improved by addressing the following issues. Therefore, I would like to reconsider the manuscript for publication in Nature Communications only after proper corrections are made.

1) There are not enough grounds to use the membrane proposed in this study instead of conventional ion-exchange membranes. Please explain in detail the superiority of the proposed membrane compared to the conventional polymer membranes.

2) It is very interesting that hybrid membranes have higher power generation performance than SNF membranes. However, it is not clear why the hybrid membrane has higher power generation performance than SNF membrane. Please give a more detailed explanation.

3) The AAO membrane is very brittle and is difficult to handle. I would like to ask if it is impossible to use other soft membranes instead of AAO membranes.

Response to Reviewer #1

Comments: *This is a nice manuscript reporting the application of silk nanofiber-based membranes for osmotic energy conversion. The porous charged silk membrane standing on anodic alumina oxide functioned as the junction for selective ion transport between concentrated and less concentrated water, converting thermodynamic salinity gradient into energy. The topic of this manuscript is timely and interesting, shows the new way of harvesting energy with easily available ion-selective membrane. The membrane, in comparison with others studied previously by others or the same group, is ease of preparation and highly stable, with possibility of large-scale production. The performance of this osmotic energy conversion system bring a significant improvement with a maximum power density up to 2.86 W/m². The authors also carried out a complete study with both experimental and simulations depth. This work is publishable by the journal, but only after the authors fully address the following comments.*

Comment 1. *I do have a serious reservation on the model of numerical simulation the authors used, which I believe is completely incorrect. The silk fiber membrane consists of tortuous pores or channels but considered as straight, this is not scientifically sound or reliable. Even though this simplification is acceptable, the 2D-plane model should be definitely replaced with 2D-axisymmetric model or 3D model. In 2D-plane model, the model geometry is not a cylinder, but a cuboid with a length of 1 m (the default parameter of COMSOL software). The use of correct model is compulsory, and then compare the correct simulation with experiment.*

Response: We appreciate the reviewer very much for the valuable suggestions. According to the comments of reviewer, we establish a new 3D model for the current system (Figure R1).

Figure R1. The new 3D model for the SNF/AAO hybrid membrane (drawing not to scale).

The numerical simulation was performed by employing the PNP equations. In Figure R2, the hybrid membrane, Model (3), shows the remarkable ion concentration enrichment at positive voltage bias (+1 V), but Model (1) and Model (2), two symmetric nanochannels, present no ionic enrichment. These results are in accordance with the experimental measurements and confirm the superior ion transport behaviors in the hybrid system. Additionally, we have made the surface integral of the two separate membranes Model (1) and (2), and the resulted ionic concentration of AAO and SNF channels (Figure R3, $C_{(AAO_surface)}$: $\sim 4.0E-12$ mol/m, $C_{(SNF_surface)}$: $\sim 7.5E-13$ mol/m) agree well with the trend of experimental results. The new simulation models and results have

been added in the revised manuscript.

Figure R2. The 3D models of the separated membranes and hybrid membrane and corresponding ionic concentration distributions.

Figure R3. The surface integral results of the two separated membranes (a. SNF membrane; b. AAO membrane).

Comment 2. One of key issues in this system is how to balance the ion selectivity and mass flux, in which surface charges play the decisive role. The authors assigned the charge density to be 0.024 C/m^2 . However, I did not find the relevant characterization of surface charge state or how they got this value. The value is also very important for simulation.

Response: Thanks for the reviewer's comments. In the last version of the manuscript, we installed the surface charge density and used the value of 0.024 C/m^2 to obtain the affordable calculations. Such a simplification or setup is also reported to obtain a simple system (Gao, J. *et al. J. Am. Chem. Soc.* **2014**, *136*, 12265-12272; Zhu, X. *et al. Sci. Adv.* **2018**, *4*, eaau1665). In the new established 3D models, the charge density of AAO was assigned to be 0.08 C/m^2 (Yang, P. *et al. Nano Lett.* **2009**, *9*, 3820-3825; Ntalikwa, J. W. *Bull. Chem. Soc. Ethiop.* **2007**, *21*, 117-128; Zhang, Z. *et al. Adv. Mater.* **2016**, *28*, 144-150). Also, the charge density of SNF was calculated according to the reported equations (Schaep, J. & Vandecasteele, C. *J. Membr. Sci.* **2001**, *188*, 129-136):

$$\sigma = \frac{\varepsilon\varepsilon_0\zeta}{\lambda_D}$$

$$\lambda_D = \sqrt{\frac{\varepsilon\varepsilon_0RT}{2n_{bulk}z^2F^2}}$$

where ϵ , ϵ_0 , n_{bulk} , and z were the permittivity of water, the permittivity of a vacuum, the concentration of solution, and the valence number, respectively. F , T , R and ζ represented the Faraday's constant, the absolute temperature, the universal gas constant, and the zeta potential of the membrane, respectively. According to the above equations and the measured zeta potential of SNF (Figure R4), the surface charge density of SNF was calculated to be 0.0062 ± 0.0004 C/m². Finally, these data were employed for the simulation with new model. The relevant description and data have been added in the revised manuscript.

Figure R4. Zeta potential of SNF in 0.1 M KCl with three measured results.

Comment 3. *The authors should describe much more details of experiments. For instances, how they measure E_{redox} , and specify clearly the experimental membrane area in this energy conversion system, as well as the area they used for estimating the conversion efficiency.*

Response: Thanks for the reviewer's comments. In the experiments, the redox potential is stemmed from the unequal potential drop at the electrode-solution interface in electrolytes with different concentrations. The redox potential is obtained by measuring the potential difference without the hybrid membrane. In all the measurements, the effective experimental area is about 3×10^{-2} mm². The corresponding discussion and data have been added into the parts of Electrical Measurements and Potential Calibration in the revised manuscript.

Comment 4. *Line 224, "The corresponding inner resistance ($r_{channel}$), which was calculated by $r_{channel} = V_{OC}/I_{SC}$...". This resistance also includes the solution resistance.*

Response: We appreciate the reviewer very much for the comments. As shown in Figure R5, the measured V_{OC} actually consists of two parts: the diffusion potential (E_{diff}) which is contributed by the power source and the redox potential (E_{redox}). The E_{redox} obtained in the electrode calibration process has already include the contribution of the solution and the electrode-solution interface. Thus, the resistance has already been subtracted via an electrode calibration process. Therefore, the equation, $r_{channel} = V_{OC}/I_{SC}$, is employed to obtain the corresponding inner resistance (Radenovic, A. *et al. Nature* **2016**, 536, 197-200; Gao, J. *et al. J. Am. Chem. Soc.* **2014**, 136, 12265-12272).

Figure R5. Equivalent circuit diagram of the power source. The measured V_{OC} is composed of two parts, E_{redox} and E_{diff} .

Response to Reviewer #2

Comments: *The manuscript from Wen et al. presented the fabrication and the performance of a silk-AAO hybrid membrane system for harvesting osmotic energy. The membrane was prepared through easy filtrating silk nanofibers with alumina membranes and presented excellent energy conversion performance. The energy conversion of the hybrid system was investigated in detail by considering a series of factors, such as ionic concentration gradients, the opening diameter of the alumina part, and thickness of the silk part of the system, which provided systematical guidance for membrane design and speeding up the development of the current filed. Also, the factor analysis was clearly conducted by employing the theoretical simulation. The presented system was scalable, easily accessible, and of potential interdisciplinary interests. The data is solid and well presented with high-quality pictures. In the meantime, this work will definitely arouse the interest of the membrane-based energy conversion with a series of low-cost materials and draw much attention from not only the academic but also the industrial community. Overall, I would like to recommend this work to be published in Nature Communications after the authors consider the following minor suggestions.*

Comment 1. *Normally, the resistance of the membrane for RED is a key factor for energy conversion performance. The current membrane showed low resistance which could benefit the energy conversion process. The fig. 3i shows the resistance increase as the concentration gradient increasing and supplementary fig. 10 shows the energy conversion efficiency decreased with the salinity gradient increasing. Thus, the authors are recommended to discuss how the concentration gradient affected the output power performance in detail for readers' interest.*

Response: We appreciate the reviewer very much for the comments. According to the equations:

$$\eta_{\max} = \frac{1}{2}(2t_n - 1)^2$$

where t_n represents the anion transference number. Thus, the η_{\max} and the square of t_n are positive correlation.

$$E_{\text{diff}} = (2t_n - 1) \frac{RT}{zF} \ln \frac{\gamma_{C_H} C_H}{\gamma_{C_L} C_L}$$

R , T , z , F , and γ are the gas constant, temperature, charge number, Faraday constant, and mean activity coefficient, respectively. C_H and C_L represent high concentration and low concentration, respectively. Here, the t_n increases with a decrease of the concentration of either side in the nano-sized channel. (Kim, D.-K. *et al. Microfluid. Nanofluid.* **2010**, 9, 1215-1224; Karnik, R. *et al. Nano Lett.* **2005**, 5,1638-1642; Stein, D. *et al. Phys. Rev. Lett.* **2004**, 93, 035901). Therefore, along with the concentration increasing, the t_n decreases, leading to the η_{\max} decreases. The corresponding discussion has been added in the revised manuscript.

Comment 2. For the AAO channel size screening part, the authors used the effective pore overlapping for analyzing the output power generation. Here, the AAO layer and silk layer are simplified to round pores with the proper size from experimental results. It's an interesting method for the discussion and easy to understand. I have noticed that the randomly generated silk layer pores distributed differently in supplementary figure 12a and 12b. Thus, I wonder how the authors generate randomly pore distribution? Also, should the generated pore distribution different in the comparison or might the authors employ more generated random silk layer pore arrays for the analysis?

Response: Thanks for the reviewer's comments. In the experiments, we generated the randomly spot distribution by employing the software, *Matlab 2010*, with the following codes:

```
x=rand(300,2)
plot(x(:,1),x(:,2),'*')
```

Figure R6. The random spots generated by employing the softwares *Matlab 2010* and *Origin 2016*.

The above Figure R6 is generated by using the software *Origin 2016*, and the corresponding data is provided by *Matlab 2010*. Different random spots can be obtained by modifying relevant parameters as required. The quantitative relationship between the SNF and AAO channels can be determined according to the proportion of diameters of the two channels.

Comment 3. In the manuscript, the authors adjusted the pH of the solution for the RED use. As known that the energy conversion is benefited from SNF/AAO heterogeneous structure, especially the opposite charge polarity in the alkaline condition. Thus, could the authors comment on the potential methods to adjust the optimal membrane for certain applications or the direction of the heterogeneous membrane development.

Response: We appreciate the reviewer very much for the comments. In view of the practical applications, the current hybrid membrane shows high performance in alkaline solutions and might be used in industrial wastewater, which is usually alkaline.

If the application is for the acidic solutions, then, the heterogeneous membrane could be developed by constructing the similar heterogeneous systems with opposite charge distributions. Besides, the stable membrane in extreme environments is also an important direction for the membrane developing.

Comment 4. *For the practical applications, the provided data table in supplementary materials helps a lot for readers' information. Also, the analysis related to fig. 4 and fig. 5 may help promote the performance of the system in practical use. The authors are recommended to comment on the practical use of the current system and the concerns or factors for developing future systems.*

Response: We appreciate the reviewer very much for the comments. Through the experimental results, we can find that the hybrid membrane achieved the best performance in the alkaline solution. Therefore, the hybrid membrane can be mainly used in alkaline salt solution for the practical applications. For developing future systems, the introduction of asymmetric factors, such as structure, charge polarity, and wettability, could guide the high performance system design.

Response to Reviewer #3

Comments: *The authors investigated silk-based hybrid membranes for reverse electrodialysis. This paper showed interesting results regarding ion transport phenomena in hybrid membranes for reverse electrodialysis. However, I believe that the quality of the paper can be further improved by addressing the following issues. Therefore, I would like to reconsider the manuscript for publication in Nature Communications only after proper corrections are made.*

Comment 1. *There are not enough grounds to use the membrane proposed in this study instead of conventional ion-exchange membranes. Please explain in detail the superiority of the proposed membrane compared to the conventional polymer membranes.*

Response: We appreciate the reviewer very much for the comments. For comparison, four types of commercially available cation-exchange membranes including Ionsep, Nafion-110, CMI, and FKS membranes (Gao, J. *et al. J. Am. Chem. Soc.* **2014**, 136, 12265-12272), and several conventional polymer membranes, such as polycarbonate (PC) membrane (Kwon, K. *et al. Int. J. Energy Res.* **2014**, 38, 530-537), polyether sulfone/sulfonated polyether sulfone (PES/SPES) membrane (Huang, X. *et al. Nano Energy* **2019**, 59, 354-362), PES-Py/PAEK-HS (Zhu *et al. Sci. Adv.* **2018**, 4, eaau1665) membrane, and BCP (Zhang, Z. *et al. J. Am. Chem. Soc.* **2017**, 139, 8905-8914) membranes were compared and the fabricated SNF/AAO membrane did show the superiority performances. The corresponding results were listed in Table R1 as follows:

Table R1. Comparison with conventional ion-exchange membranes.

Membrane type	U_{oc} (mV)	I_{sc} (A/m ²)	Thickness (μ m)	P_{max} (W/m ²)	Concentration gradient
Ionsep ¹	99.6	15.3	540	0.37	50
Nafion-110 ¹	99.0	13.2	20	0.33	50
CMI ¹	99.1	14.2	320	0.40	50
FKS ¹	100.2	12.5	20	0.26	50
PC ²	55	4.7	20	0.058	1000
PES/SPES ³	/	/	91	2.48	500
PES-Py/PAEK-HS ⁴	/	/	11	2.66	50
BCP ⁵	/	/	0.5	2.1	50
SNF/AAO ⁶	58	135	75	2.86	50

[1] Gao, J. *et al. J. Am. Chem. Soc.* **2014**, 136, 12265-12272. [2] Kwon, K. *et al. Int. J. Energy Res.* **2014**, 38, 530-537. [3] Huang, X. *et al. Nano Energy* **2019**, 59, 354-362. [4] Zhu *et al. Sci. Adv.* **2018**, 4, eaau1665. [5] Zhang, Z. *et al. J. Am. Chem. Soc.* **2017**, 139, 8905-8914. [6] The current work.

In classic ion-exchange membranes, the size of the ionic species is comparable to the channel width (typically less than 1 nm), so that the ions transport through such membrane channels encountering great steric hindrance, resulting in low ionic conductivity (Okada, T. *et al. Electrochim. Acta* **1998**, 43, 3741-3747). Moreover, the asymmetric pore structure and the bipolar charge distribution help to suppress the concentration polarization, especially at the low-concentration side (Długołęcki, P. *et al. Environ. Sci. Technol.* **2009**, 43, 6888-6894). These functions can not be realized in the conventional ion-exchange membranes, such as those listed in the table, were not of these functions. Therefore, owing to the high ionic flux (large open channels) and high short-circuit current which is one order higher than that of the conventional

ion-exchange membranes, the SNF/AAO membrane shows the superior power output for the energy conversion.

Comment 2. *It is very interesting that hybrid membranes have higher power generation performance than SNF membranes. However, it is not clear why the hybrid membrane has higher power generation performance than SNF membrane. Please give a more detailed explanation.*

Response: Thanks for the reviewer's comments. The higher power generation performance of hybrid membrane benefits from the introduction of the AAO channel with the ion-storage function, which could largely induce the ion enrichment in the channels and show the resultant bigger current than that of the SNF membrane. As explained in our response to comment 1, the asymmetric pore structure and the bipolar charge distribution help to suppress the concentration polarization (Długołęcki, P. *et al. Environ. Sci. Technol.* **2009**, *43*, 6888-6894). These descriptions are shown in Figure R7 (which is used in the manuscript as Fig. 3d), also, we have added some detailed explanations in the revised manuscript according to the comment of Reviewer 3.

Figure R7. Schematics of the AAO/SNF structure. The AAO and SNF membranes are considered to be an ion-storage layer and ion-selection layer, respectively.

Comment 3. *The AAO membrane is very brittle and is difficult to handle. I would like to ask if it is impossible to use other soft membranes instead of AAO membranes.*

Response: We appreciate the reviewer very much for the comments. The AAO membranes are chosen in the current work for several reasons. First, AAO membrane is the most used substrate material due to its controllable channel structures, accessible surface polarity tuning, and stability, which have widely applied as templates and substrates in various fields, especially, in nanofluidic systems (Zhang, Z. *et al. Adv. Mater.* **2016**, *28*, 144-150). Second, the volume production and mature technology of AAO membrane lay the foundation for the large-scale application of hybrid membrane. In addition, we have also employed soft PC membranes to hybridize with SNF membrane, and the SNF/PC showed a low output power density. We believe that the abundant interface groups (-OH) of AAO could enhance ions transport through hybrid membrane. Besides, we are currently striving for other soft membranes aiming to get materials with fascinating energy conversion performance.

REVIEWERS' COMMENTS:

Reviewer #1 (Remarks to the Author):

The authors have seriously addressed all my comments and revised their manuscript. In particular, I am very pleased to see their modification of Comsol simulations with a three dimensional model, which is correct and accurate. I would recommend the publication of the manuscript by the journal.

Reviewer #2 (Remarks to the Author):

The authors have addressed the suggestions and concerns in detail. The using of software to generate random pore arrays is convinced, and the discussions about designing and developing heterogeneous systems could also raise the readers' interests. The revised manuscript is suitable for publication.

Reviewer #3 (Remarks to the Author):

I recommend the paper to be accepted for published in Nature Communications. The authors carefully accomplished additional clarifications and modifications which I recommended in the previous review.

REVIEWERS' COMMENTS:

Reviewer #1 (Remarks to the Author):

The authors have seriously addressed all my comments and revised their manuscript. In particular, I am very pleased to see their modification of Comsol simulations with a three dimensional model, which is correct and accurate. I would recommend the publication of the manuscript by the journal.

Response: We appreciate the reviewer very much for the comments.

Reviewer #2 (Remarks to the Author):

The authors have addressed the suggestions and concerns in detail. The using of software to generate random pore arrays is convinced, and the discussions about designing and developing heterogeneous systems could also raise the readers' interests. The revised manuscript is suitable for publication.

Response: Thank the reviewer for the comments.

Reviewer #3 (Remarks to the Author):

I recommend the paper to be accepted for published in Nature Communications.
The authors carefully accomplished additional clarifications and modifications which I recommended in the previous review.

Response: We appreciate the reviewer very much for the recommendation.